

# Comprehensive analysis of transcriptomics and radiomics revealed the potential of TEDC2 as a diagnostic marker for lung adenocarcinoma

Qian Huang[1],[*], Peng Zhang[2],[*], Zhixu Guo[3], Min Li[4], Chao Tao[2] and Zongyang Yu[5]

[1] Department of Hepatobiliary Disease, The Third People's Hospital of Fujian University of Traditional Chinese Medicine, Fuzhou, China
[2] Radiodiagnostic Department, The 900 Hospital of the Joint Service Support Force of the People's Liberation Army of China, Fuzhou, China
[3] Information Department, The 900 Hospital of the Joint Service Support Force of the People's Liberation Army of China, Fuzhou, China
[4] Pathology Department, The 900 Hospital of the Joint Service Support Force of the People's Liberation Army of China, Fuzhou, China
[5] Respiratory Department, The 900 Hospital of the Joint Service Support Force of the People's Liberation Army of China, Fuzhou, China
[*] These authors contributed equally to this work.

Corresponding authors
Chao Tao, taocc1986@sina.cn
Zongyang Yu, yuzy527@sina.com

## ABSTRACT

**Background:** Lung adenocarcinoma (LUAD) is a widely occurring cancer with a high death rate. Radiomics, as a high-throughput method, has a wide range of applications in different aspects of the management of multiple cancers. However, the molecular mechanism of LUAD by combining transcriptomics and radiomics in order to probe LUAD remains unclear.

**Methods:** The transcriptome data and radiomics features of LUAD were extracted from the public database. Subsequently, we used weighted gene co-expression network analysis (WGCNA) and a series of machine learning algorithms including Random Forest (RF), Least Absolute Shrinkage and Selection Operator (LASSO) logistic regression, and Support Vector Machines Recursive Feature Elimination (SVM-RFE) to proceed with the screening of diagnostic genes for LUAD. In addition, the CIBERSORT and ESTIMATE algorithms were utilized to assess the association of these genes with immune profiles. The LASSO algorithm further identified the features most relevant to the expression levels of LUAD diagnostic genes and validated the model based on receiver operating characteristic (ROC), precision-recall (PR), calibration curves and decision curve analysis (DCA) curves. Finally, RT-qPCR, transwell and cell counting kit-8 (CCK8) based assays were performed to assess the expression levels and potential functions of the screened genes in LUAD cell lines.

**Results:** We screened a total of 214 modular genes with the highest correlation with LUAD samples based on WGCNA, of which 192 genes were shown to be highly expressed in LUAD patients. Subsequently, three machine learning algorithms identified a total of four genes, including UBE2T, TEDC2, RCC1, and FAM136A, as diagnostic molecules for LUAD, and the ROC curves showed that these diagnostic molecules had good diagnostic performance (AUC values of 0.989, 0.989, 989, and

0.987, respectively). The expression of these diagnostic molecules was significantly higher in tumor samples than in normal para-cancerous tissue samples and also correlated significantly and negatively with stromal and immune scores. Specifically, we also constructed a model based on TEDC2 expression consisting of seven radiomic features. Among them, the ROC and PR curves showed that the model had an AUC value of up to 0.96, respectively. Knockdown of TEDC2 slowed down the proliferation, migration and invasion efficiency of LUAD cell lines.

**Conclusion:** In this study, we screened for diagnostic markers of LUAD and developed a non-invasive radiomics model by innovatively combining transcriptomics and radiomics data. These findings contribute to our understanding of LUAD biology and offer potential avenues for further exploration in clinical practice.

# INTRODUCTION

Lung cancer is a frequent cancer, and its survival rate is critically dependent on the stage at diagnosis (*Siegel et al., 2023*; *Yu et al., 2023b*; *Ding, Lv & Hua, 2022*). Patients at stage IIIa-IVA of lung cancer normally have a 5-year survival rate ranging from 10–6%, while that of patients with stage I reaches 7–92% (*Chinese Thoracic Society, 2023*; *Yu et al., 2023a*). Thanks to progress in diagnosis, surgical techniques, radiotherapy, and molecular therapies, the clinical prognosis for patients with lung adenocarcinoma (LUAD) has markedly enhanced. Nevertheless, the 5-year survival rate for individuals with LUAD remains significantly low (*Zhang et al., 2019*; *Jurisic et al., 2020*; *Mao et al., 2020*). This could be attributed to patients being diagnosed at advanced stages, or to early-stage patients not being eligible for targeted therapy due to the absence of common molecular mutations such as EGFR, BRAF V600E, MET, or ALK (*Feng et al., 2022*). Therefore, further research into the molecular mechanisms of tumorigenesis and the development of new, reliable biomarkers is essential to enhance the survival outcomes for LUAD patients.

Radiomics is a high-throughput method for extracting quantitative features from standard medical imaging (*Lambin et al., 2017*). It thoroughly examines image properties and employs sophisticated statistical methods to determine the features most closely linked to clinical results. This technique builds on extensive research in computer-aided diagnosis and pattern recognition (*Fornacon-Wood et al., 2020*). Compared to traditional tissue sampling methods, radiomics offers several advantages: it is non-invasive, reproducible, cost-effective, and less susceptible to the variability caused by intratumoral heterogeneity (*Wang et al., 2024*; *Pan et al., 2023*; *Chen et al., 2023*). As a result, radiomics has broad applications in various aspects of cancer diagnosis and treatment, though further validation is required before it can be widely implemented in clinical practice. Currently, genomics of radiation research has concentrated on identifying and linking known biological characteristics, including isocitrate dehydrogenase-1 (IDH-1), the epidermal

growth factor receptor (EGFR), P53 mutations, BRCA1/2, Kirsten rat sarcoma (KRAS), BRCA1-associated protein 1 (BAP1), as well as other genetic mutations and molecular subtypes (*Kim et al., 2020*; *Zhang et al., 2020*; *Li et al., 2018*; *Gierach et al., 2014*; *Kang et al., 2023*). As expected, radiomics has been developed to predict pathological relevance in lung cancer (*Fatima, Jaiswal & Sachdeva, 2022*). There have been studies that have explored the combination of radiomics and transcriptomics, with some researchers created a radiotranscriptomic signature by utilizing serum miRNA levels and CT texture features to anticipate how patients with non-small cell lung cancer (NSCLC) will respond to radiotherapy. This special signature has the potential to function as a stand-alone biomarker in assessing the efficacy of radiation therapy for NSCLC patients (*Fan et al., 2020*). However, there is a lack of screening for radiomic biomarkers predictive of LUAD patients to provide diagnostic and therapeutic value for cancer intervention.

In this study, we took the expression profile and radiomics features of LUAD as the starting point, used multiple machine learning analysis methods to identify biomarkers for the diagnosis, prognosis monitoring and tumor immunology of LUAD, and developed a radiomics model of LUAD for non-invasive testing of biomarkers. This innovatively combines imaging histology and transcriptomics data, which not only provides a comprehensive picture of tumor characteristics, but also significantly improves the accuracy and clinical application potential of LUAD early detection.

# MATERIALS AND METHODS

## Selection and processing of transcriptomics cohorts

The TCGA-LUAD cohort was selected from the TCGA database (https://portal.gdc.cancer.gov) to acquire transcriptomics data and clinicopathological information in FPKM form, and a total of 572 samples (including 513 tumor samples and 59 para-cancerous tissue samples) were included in this cohort. In this research, the RNA-sequencing data from TCGA was transformed into transcripts per kilobase million (TPM) values, facilitating better comparability between TCGA samples and microarray datasets (*Wagner, Kin & Lynch, 2012*). Transcriptome data of LUAD samples were obtained from the Gene Expression Omnibus (GEO, https://portal.gdc.cancer.gov) database using the search numbers GSE31210 and GSE30219. The dataset numbered GSE31210 included 20 normal samples and 226 tumor samples. The cohort with search number GSE30219 included 14 normal samples and 293 tumor samples. The downloaded data in the GEO database were processed by the R package oligo (*Carvalho & Irizarry, 2010*) according to the uniform data preprocessing routine.

## Construction of co-expression networks

Weighted gene co-expression network analysis (WGCNA) identifies gene modules that are significantly associated with characterized phenotypes by constructing gene co-expression networks, which helps us to identify a set of potential candidate genes that are most relevant to LUAD (*Langfelder & Horvath, 2008*). Thus, in this study, based on the characteristic that gene co-expression analysis is sensitive to abnormal values, the Median Absolute Deviation (MAD) of all protein coding genes in the whole genome was

calculated, and the genes with MAD greater than the top 70% were submitted to the "WGCNA" package (*Langfelder & Horvath, 2008*) for weighted gene co-expression network development. The distance-based adjacency index of the sample was calculated and the default parameters are defined to generate the module. After cluster analysis of modules, a heatmap of correlation between modules and traits was constructed.

## Pathway annotation analysis

The data of the two queues downloaded from the GEO database were merged, and the RNA data of LUAD samples and normal samples were submitted to the R package "limma" (*Ritchie et al., 2015*) for difference analysis. The threshold of differentially expressed genes (DEGs) was defined as adj. $p < 0.05$ and | log2FC | > 1. Gene Ontology (GO) and Kyoto Encyclopedia of Genes and Genomes (KEGG) analysis was performed by importing the DEGs into the "clusterProfiler" package (*Wu et al., 2021*). The complete expression profile was loaded into clusterProfiler package for gene set enrichment analysis (GSEA). The results of GO and KEGG analysis were visualized as bars by ggplot2 package. GSEA enrichment maps were generated by gseaplot2.

## Machine learning analysis

Machine learning analysis was performed to select diagnostic markers for LUAD, including Random Forest (RF), Least Absolute Shrinkage and Selection Operator (LASSO) logistic regression and Support Vector Machines Recursive Feature Elimination (SVM-RFE). RF is an integrated learning method based on decision trees that efficiently handles high-dimensional data and provides an importance score for each feature. With the importance score of RF, we can filter out the genes with the most discriminative ability in the classification task, reduce the spatial dimensionality of features, and provide the generalization ability of the model (*Hu & Szymczak, 2023*). In this study, the 'randomForest' package in R was used to grow a forest of 610 trees using the default settings (*Alderden et al., 2018*). The glmnet function of "glmnet" package was used to perform the LASSO Cox regression model analysis, in which the parameters were family = "binomial", alpha = 1, and nlambda = 100 (*Engebretsen & Bohlin, 2019*). The LASSO algorithm is able to select a small number of important features while avoiding overfit and to further improve the stability and prediction ability of the model (*Jiang & Jiang, 2023*). In addition, in the SVM-RFE, the SVM classifier was constructed using R package e1071, which the parameters were kernel = "linear", and "cost = 1". This is due to the ability of the SVM-RFE method to finely select features through a recursive process to ensure that the selected features have the maximum contribution to the model performance and thus optimize the feature set (*Sanz et al., 2018*). The genes jointly selected by the above machine learning algorithms were regarded as diagnostic markers of LUAD.

## Analysis of immune system characteristics

We used the CIBERSORT and ESTIMATE algorithms, respectively, to further evaluate the association between the screened diagnostic markers and the tumor microenvironment. The documents required for CIBERSORT analysis were prepared in advance, including the

official LM22, gene expression matrix and CIBERSORT code (*Newman et al., 2015*). The results of CIBERSORT were analyzed by Pearson correlation analysis together with the LUAD markers. The gene expression matrix of LUAD was converted into a GCT format file and read into ESTIMATE (*Yoshihara et al., 2013*). The immune, stromal and ESTIMATE scores of the sample were calculated by the estimateScore() function. Pearson correlation analysis was also performed to calculate the correlation coefficient between each score and each diagnostic marker screened.

## Extraction of radiomics feature

Three regions of interest (ROIs) of gross tumor volume from TCGA-LUAD were manually segmented using ITK-SNAP (version 3.8.0) by a radiologist with more than 5 years of experience. A total of 107 radiomic features: including 14 shape features, 18 firstorder statistical features, and 75 texture features (including 16 Gray level size zone matrix features, 14 Gray level dependence matrix features, 24 Gray level cooccurrence matrix features, 16 Gray level run length matrix features, and five neighbouring gray tone difference matrix features), were collected from the ROIs based on CT images applying PyRadiomics (version 3.0) (*van Griethuysen et al., 2017*). It should be noted that the SD of original shape Flatness and original shape Least Axis Length was 0 and were not considered in the subsequent analysis. The Z-score algorithm was used to normalize the radiomic features for further analysis.

## Construction of the radiomics model

Feature selection was realized applying LASSO regression analysis. To establish a robust binary classification model, we used the generalized linear model network (glmnet) with a binomial distribution for the logit link function. During the model training process, we set 1,000 different lambda values (nlambda = 1,000) and chose LASSO regression (alpha = 1) as the regularization method. To select the optimal regularization parameter lambda, we performed five-fold cross-validation (nfolds = 5) and used deviance as the evaluation metric. The cross-validation was carried out using the "cv.glmnet" function. Finally, we identified features related to TEDC2 expression from radiomic features and established the corresponding model. The formula for calculating radiomics score (Rad score) was obtained:

$$Rad\ score = \sum_{i=1}^{n} \beta_i \times feature_i + intercept$$

In the formula, $\beta$ refers to the feature coefficient ($\beta$), and intercept within the radiomics signature was determined according to the average value of the included models.

## Source of cells and RT-qPCR

Human LUAD cells A549 and human normal lung epithelial cells BEAS-2B were commercially purchased from The American Type Culture Collection (ATCC), thawed and cultured according to the supplied product instructions. Cells were cultivated at 37 °C in a DMEM (PM150210) medium that includes 10% fetal bovine serum (FBS), 1% glutamine, and a 1% solution of antibiotic/antifungal agents, with a controlled atmosphere
**Table 1 Primers of genes.**

| Gene | Forward primer sequence (5′-3′) | Reverse primer sequence (5′-3′) |
|---|---|---|
| UBE2T | ATCCCTCAACATCGCAACTGT | CAGCCTCTGGTAGATTATCAAGC |
| TEDC2 | ATGCACACCCAGTCCACAAG | CCGGCCTTAGTGATGCCTC |
| RCC1 | CGGTGTGATTGGACTGTTGGA | CACCAAGTGGTCGTTTCCTGA |
| FAM136A | TGCAGGGTCTCATGTTCCG | GCTCCTTACTCCCAGCATCTATT |
| GAPDH | CTGGGCTACACTGAGCACC | AAGTGGTCGTTGAGGGCAATG |

of 5% $CO_2$. Total RNA was separated using E.Z.N.A. Total RNA Kit I (Omega Bio-Tek, Norcross, USA), followed by reverse transcription of total RNA into cDNA with the use of PrimeScript RT Master Mix (TaKaRa, Japan). qRT-PCR was carried out using the SYBR Green RT-PCR kit (Vazyme, Nanjing, China). Relative mRNA expression was quantified by the $2^{-\Delta\Delta C}t$ method and normalized to GAPDH. Each primer sequence is described in Table 1.

## Transwell assay

siRNA for TEDC2 was synthesized by GenePharma (Shanghai, China) and transfected into A549 cells seeded in six-well plates using Lipofectamine 2000 according to the prescribed protocol. After transfection, 200 µL of A549 cells were transferred to the upper chamber without matrigel and the upper chamber coated with Matrigel, respectively, and complete medium containing 10% FBS was added to the lower chamber for 24 h. Crystal violet was used for cell staining for 30 min after harvesting the cells in the lower chamber. Infiltrating cells were quantified under a microscope. Cells in four random fields were intercepted, and photographs were taken using an inverted microscope and the number counted.

## Cell counting Kit-8 assay

Cell counting kit-8 (CCK8) assay for assessing cell viability of LUAD cell lines after silencing TEDC2. The A549 cells were seeded into a 96-well plate during the exponential growth phase at a density of $1 \times 10^4$ cells per well and incubated at 37 °C and 5% $CO_2$ for 48 h. Following this incubation period, 10 µL of CCK8 was added to the cell culture and incubated at 37 °C for 2 h. The measurements of optical density (OD) were taken at a wavelength of 450 nm (OD 450) using a microplate reader manufactured by Bio-Rad Laboratories Inc. The data presented are the average results from three separate experiments.

## Statistical analysis

Statistical tests were performed in R software (version 3.6.0). Cox regression analysis was run using the "survival" package and diagnostic efficiency was evaluated by generating receiver operating characteristic (ROC) curves using the "ROCR" package. Student's t-test or Wilcoxon rank-sum test was used to analyze continuous variables, Pearson correlation analysis was used to assess the correlation between variables, and log-rank test was used to

compare the differences in survival time between different groups of patients. A *p* value < 0.05 indicated statistical significance.

## RESULTS

### Identification and functional characterization of LUAD-related gene modules

WGCNA showed that stable average connectivity was achieved with a fit $R^2 = 0.85$ for the scale-free topological model, corresponding to a soft threshold of 7 (Figs. S1A–S1B). After executing WGCNA, 15 gene modules were obtained (Fig. S2C). The turquoise module has the largest pool of similarly expressed genes (Fig. S2D). The correlation between module expression profile and LUAD showed that blue was the module with the strongest correlation with LUAD, and correlation coefficient between module membership (MM) and gene significance (GS) reached 0.66, and 214 key genes in the blue module were identified using MM > 0.7 and GS > 0.4 as the screening criteria (Figs. 1A, 1B). These genes were significantly annotated in pathways, cellular components, biological processes, molecular functions associated with the cell cycle (Figs. 1C–1F). Therefore, the blue module may be the module that mediates the cell cycle of LUAD.

### Cell cycle progression was hyperactive in LUAD

A total of 2,600 genes were dysregulated in LUAD samples relative to normal samples, including 1,593 significantly down-regulated genes and 1,007 significantly up-regulated genes (Figs. S2A, S2B). These two classes of DEGs that showed opposite expression patterns in LUAD were also involved in different functional regulation, and the up-regulated genes were significantly annotated to numerous pathways regulating the cell cycle, such as chromosome segregation, DNA conformation change, and DNA replication, *etc.* (Fig. S2C). The genes with downregulated expression were significantly annotated in kidney development, regulation of angiogenesis, tissue migration, regulation of epithelial cell proliferation and migration (Fig. S2D). GSEA analysis of LUAD expression profile also revealed significant activation of DNA repair and G2M checkpoint pathway, which are involved in cell cycle regulation in LUAD (Fig. S3). Therefore, based on these results, we suggest that hyperactivity of the cell cycle may be a major feature of LUAD.

### Screening, validation and accuracy assessment of diagnostic markers

We found that 192 more of the 214 key genes identified in WGCNA-based were upregulated in LUAD patients (Fig. 2A). To further explore the impact of these key genes in LUAD, we first identified 18 characterized genes using the SVM-RFE algorithm (Fig. 2B). Subsequently, we identified six genes based on the RF algorithm (Fig. 2C) as well as screened 11 genes using the LASSO logistic regression method (Fig. 2D). We utilized a Wayne diagram in order to take the intersection of the genes screened by the three machine algorithms and ended up with four key genes, UBE2T, TEDC2, RCC1, and FAM136A for subsequent in-depth studies (Fig. 3A). As shown in Fig. 3D, we found that all four genes were significantly overexpressed in tumor samples compared to normal samples in the TCGA training cohort. The ROC curves of UBE2T, TEDC2, RCC1 and

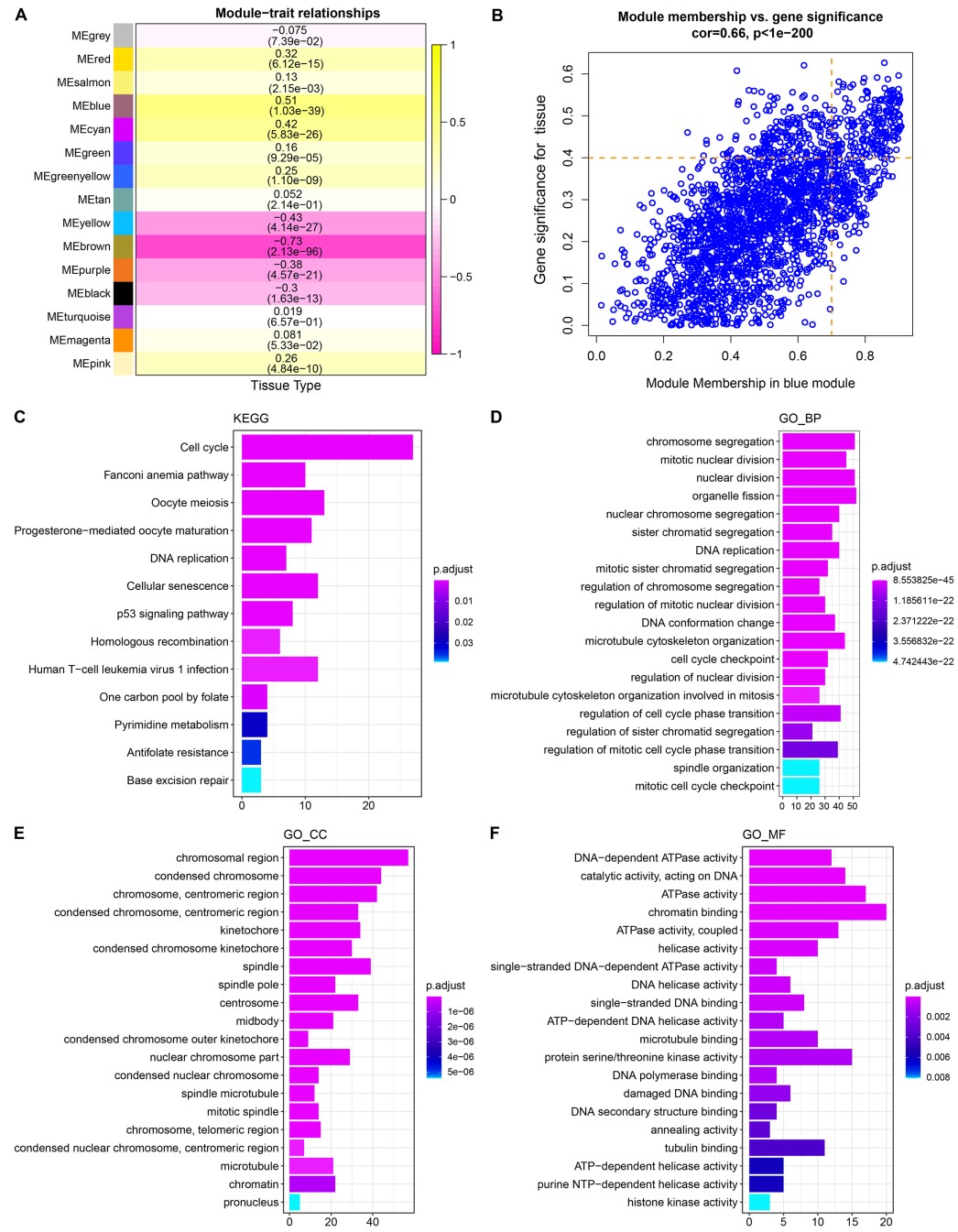

**Figure 1** **Identification and functional characterization of LUAD-related gene modules.** (A) The heatmap depicts the correlation between modules and LUAD. Each cell contains the corresponding correlation coefficient and *p*-value after correction for multiple testing. (B) Correlation between gene significance (GS) and module membership (MM). (C) Significantly annotated KEGG pathways for 214 genes in the blue module. (D–F) GO biological processes (BP), cellular components (CCs), molecular functions (MFs) that significantly annotated by 214 genes in the blue module.

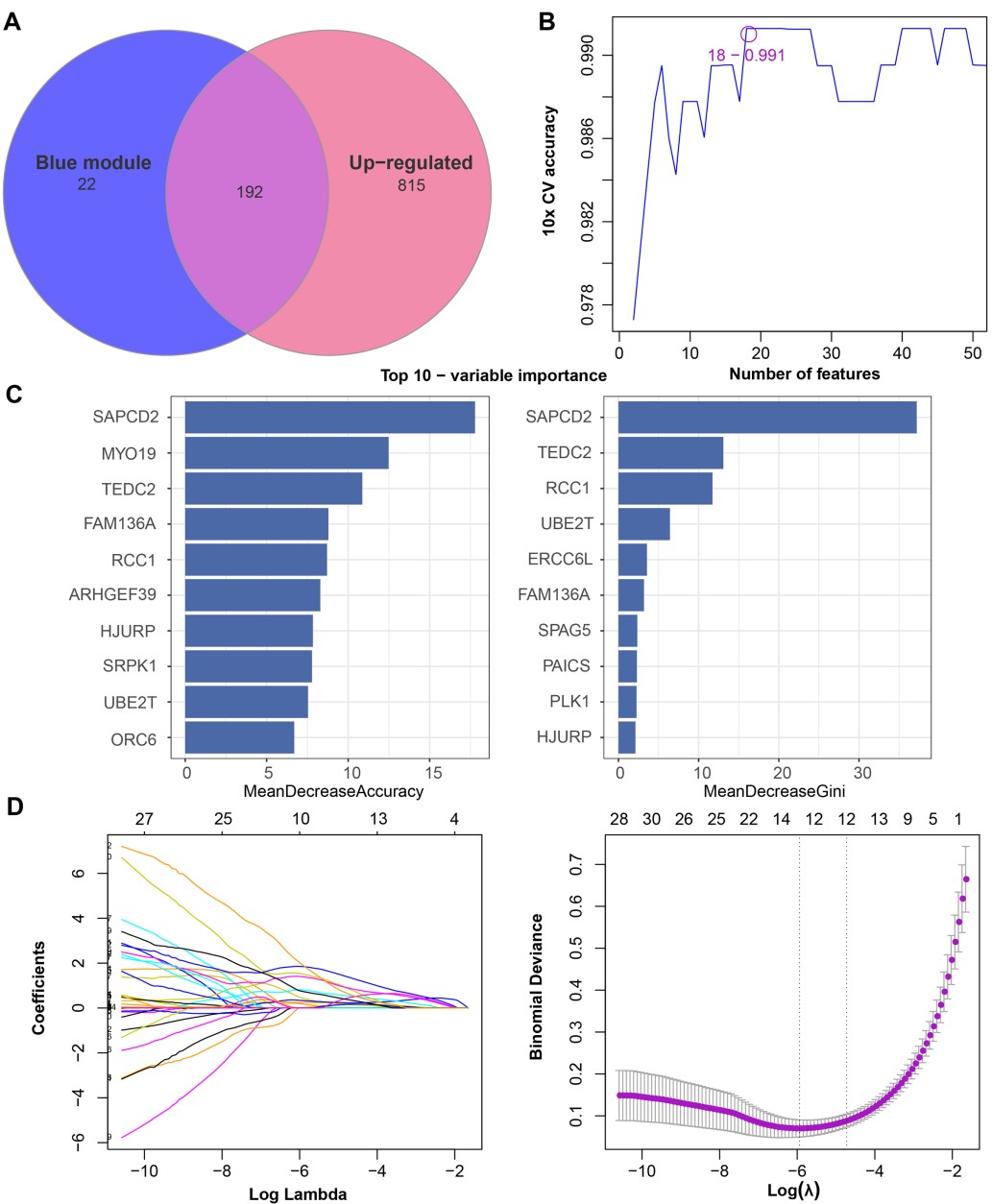

**Figure 2 Feature genes selected by different machine learning analysis methods.** (A) Coexistence analysis of key genes of the blue module with DEGs up-regulated in LUAD. (B) The variation curve of the error value predicted by different gene combinations, the abscissa represents the number of features, and the ordinate $10 \times CV$ accuracy represents the accuracy of the curve change after 10 times cross-validation. (C) The relative importance of the variables calculated by RF analysis, "mean decrease accuracy" represents the degree of decline in the accuracy of random forest prediction, and "mean decrease gini" calculates the influence of each variable on the heterogeneity of observations on each node of the classification tree, thus comparing the importance of variables. (D) LASSO logistic regression was conducted to select feature genes.

FAM136A showed their probability of being valuable biomarkers with the area under the ROC curves (AUC) value of 0.989, 0. 989, 989 and 0.987, respectively, which suggests that these four biomarkers have high predictive value (Fig. 3B). Finally, we combined these four

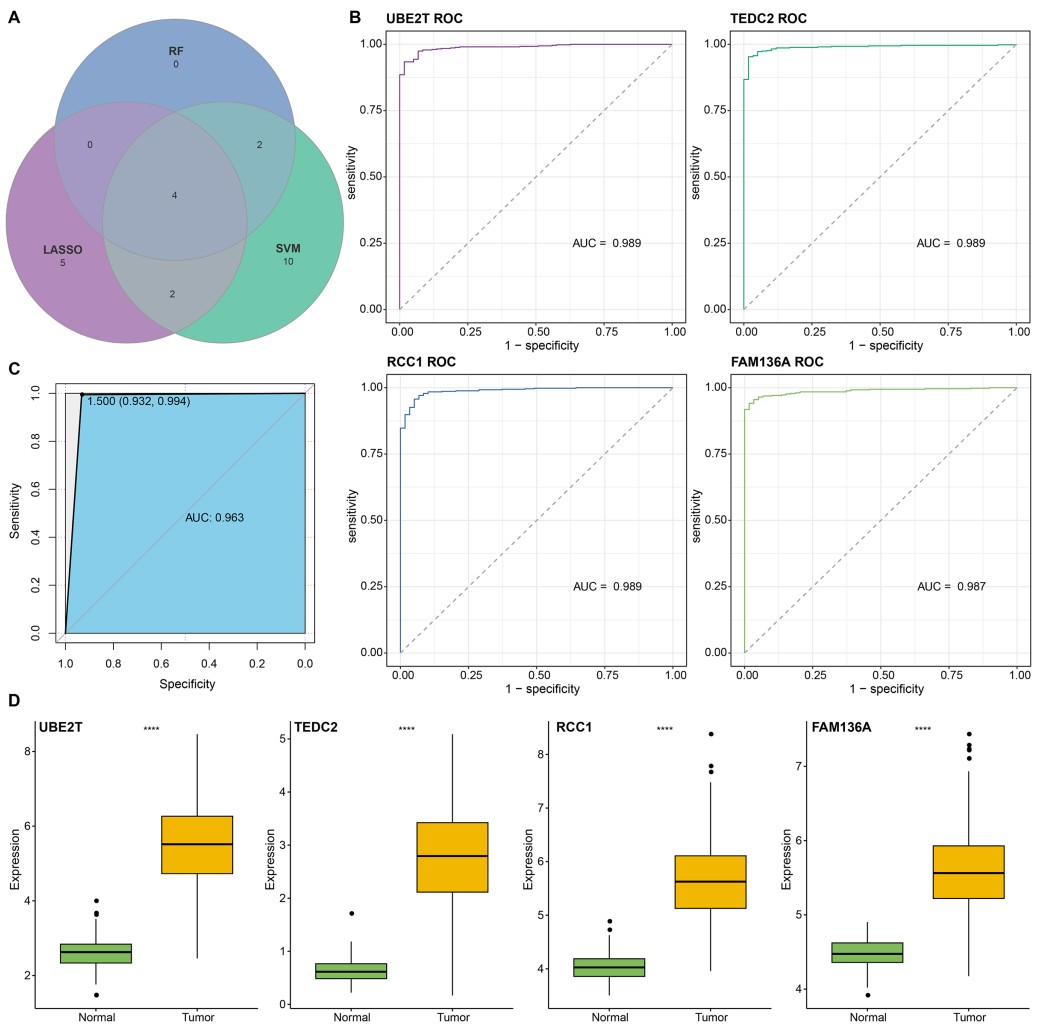

**Figure 3 Screening and accuracy assessment of diagnostic markers.** (A) Intersection genes were screened from the selected genes from SVM-RFE, RF, and LASSO regression analyses. (B) ROC curves of UBE2T, TEDC2, RCC1 and FAM136A for the diagnosis of LUAD. (C) ROC curve of logit regression model composed of four genes for the diagnosis of LUAD. (D) Based on the TCGA-LUAD dataset in order to explore the differences in expression levels of the four diagnostic genes between normal and tumor samples. **** represents $p < 0.0001$.              

biomarkers and found the AUC value of up to 0.963 based on a logit regression model, which again showed high diagnostic accuracy (Fig. 3C).

Subsequently, we further validated the screened diagnostic genes using the GSE30219 and GSE31210 datasets as validation. The AUC value of UBE2T, TEDC2, RCC1 and FAM136A as diagnostic markers in the GSE30219 cohort were 0.96, 0.95, 0.89 and 0.94, respectively (Fig. 4A). In GSE31210 cohort, the AUC value of UBE2T, TEDC2, RCC1 and FAM136A for the diagnosis of LUAD were 0.96, 0.94, 0.94 and 0.93, respectively (Fig. 4C). Additionally, all four diagnostic markers showed significantly higher expression in LUAD tumor samples than in normal samples in both cohorts (Figs. 4B, 4D). Therefore, the accuracy of the four genes as diagnostic markers in the GSE30219 and GSE31210 cohorts was also ideal.

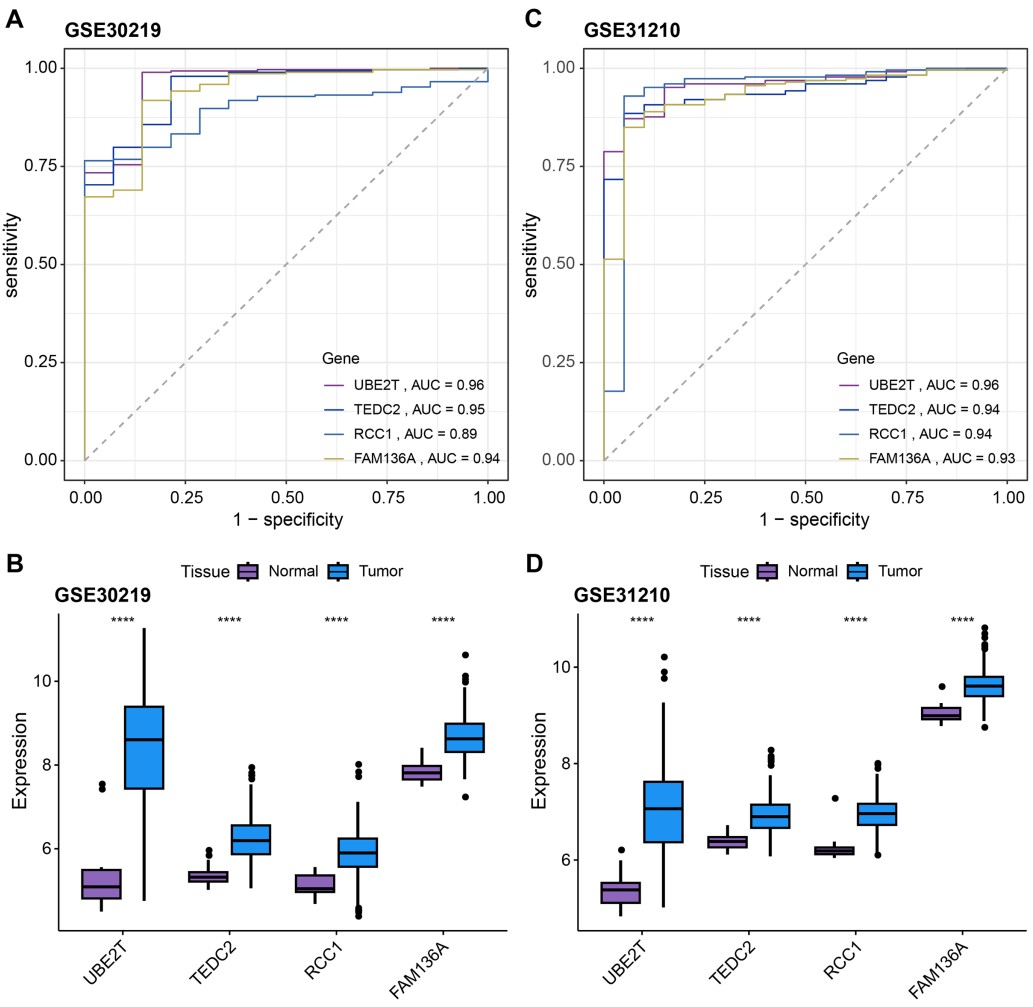

**Figure 4 Verification of diagnostic markers.** ROC curves of UBE2T, TEDC2, RCC1 and FAM136A as LUAD diagnostic markers in (A) GSE30219 cohort and (B) GSE31210 cohort. Differential expression of four diagnostic genes between tumor samples and normal samples in (C) GSE30219 cohort and (D) GSE31210 cohort. **** represents *p* < 0.0001.

## Association of diagnostic markers with LUAD prognosis and TME

In addition to showing good performance for the diagnosis of LUAD, we also explored the relationship between these four genes and patient prognosis. Using univariate Cox regression analysis, we found that UBE2T, TEDC2, RCC1 and FAM136A were significantly associated with patients' overall survival (Fig. S4A), whereas only UBE2T was significantly associated with patients' progression free interval (PFI) (Fig. S4C). Additionally, multivariate Cox regression analysis showed that these four diagnostic genes were not independent factors for predicting the prognosis of LUAD (Figs. S4B, S4D).

Next, we further explored the relationship between these four key genes and tumor microenvironment (TME). Notably, we found that UBE2T, TEDC2, RCC1 and FAM136A were all significantly and positively associated with M1 Macrophages, activated memory CD4 T cells, and follicular helper T cells, whereas they were significantly and positively associated with M2 macrophage and resting memory CD4 T cells (Figs. 5A–5D). In

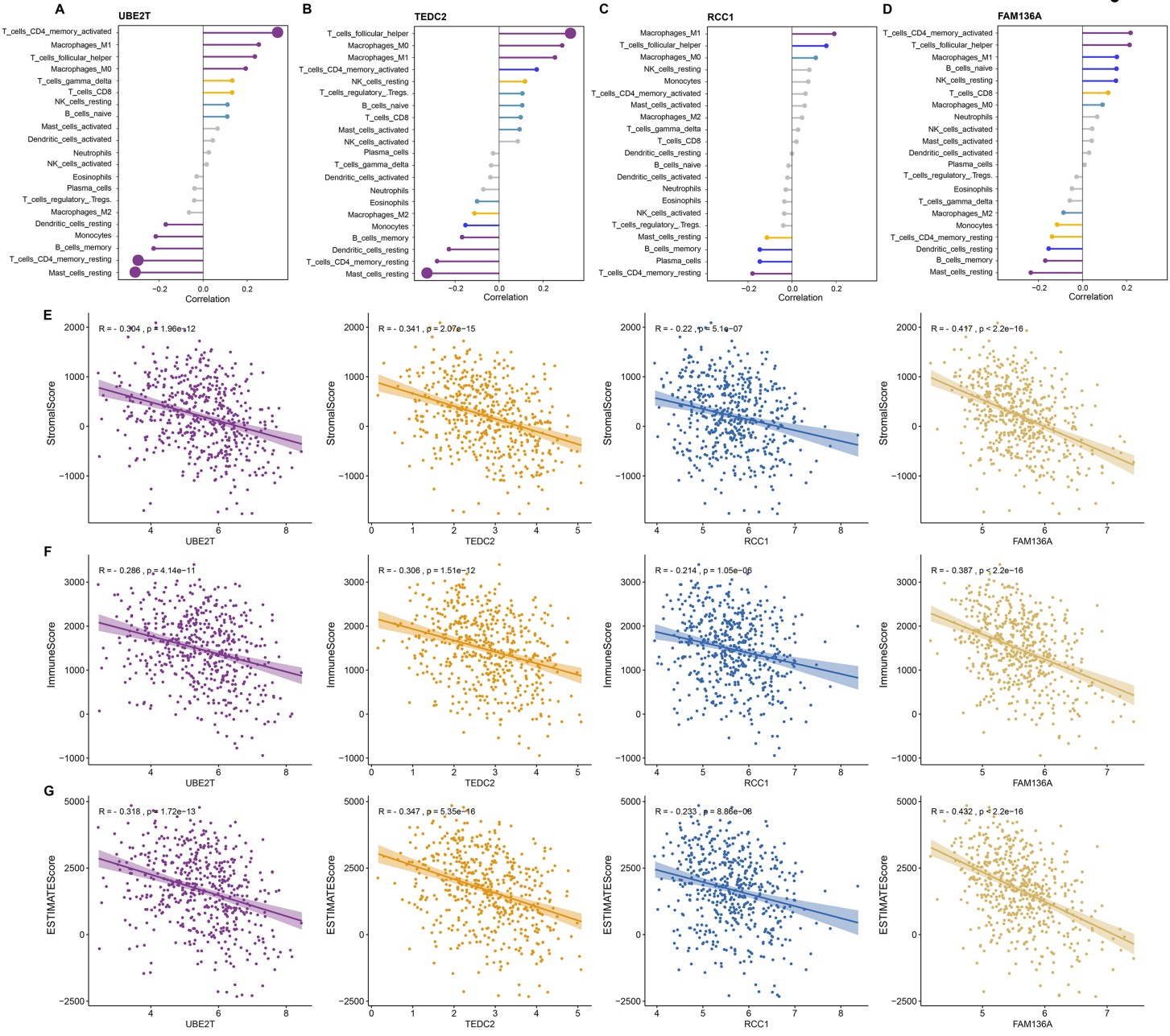

**Figure 5 Association of diagnostic markers with TME.** (A–D) The relationship between UBE2T (A), TEDC2 (B), RCC1 (C) and FAM136A (D) and immunocyte infiltration. (E–G) Pearson correlation analysis between each diagnostic gene and stromal score (E), immune score (F), EATI-MATE score (G).

addition, we found that each diagnostic gene was also negatively correlated with stromal, immune and ESTIMATE scores (Figs. 5E–5G). These results indicate that the four biomarkers we screened may act by inhibiting stromal and immune cells, which in turn promotes the malignant behavior of tumors.
## Construction and discriminant ability evaluation of radiomics model

Since we did not identify features associated with the expression levels of UBE2T, RCC1 and FAM136A. Therefore, we only analyzed the mechanism of action of TEDC2 in a follow-up study. As shown in Figs. 6A, 6B, we selected the seven optimal features associated with TEDC2 expression, including "original shape Elongation", "original firstorder Median", "original firstorder Total Energy", "original glrlm Run Length NonUniformity", "original glszm Large Area Emphasis", "original glszm Small Area Emphasis", and "original glszm Small Area Low Gray Level Emphasis", from 105 radiomics features by the LASSO algorithm. Therefore, the resulting radiomics model was: Rad score= −0.5888599−0.6561874* original shape Elongation + 2.6456259* original firstorder Median−1.1273691* original irstorder Total Energy + 0.9690754* original glrlm Run Length NonUniformity−1.4601943* original glszm Large Area Emphasis −1.4749247* original glszm Small Area Emphasis−0.9582481* original glszm Small Area Low Gray Level Emphasis.

To demonstrate the practicability of radiomics model in the diagnosis of LUAD, the sensitivity (SEN), accuracy (ACC), specificity (SPE), Positive Predictive Value (PPV) and Negative Predictive Value (NPV) scores of radiomics model were calculated by using 60% of the samples in TCGA as the training set and 40% as the verification set, which were 0.85, 0.9, 0.8, 0.8889 and 0.8182 respectively, and ROC-AUC was 0.96 (Fig. 6C). In the validation cohort, SEN, PPV, ACC, SPE, and NPV of the radiomics model were 0.8889, 1.0, 0.75, 1 and 0.8333, respectively. The ROC-AUC reached 0.9, and the Precision Recall (PR)-AUC reached 0.96 (Fig. 6D). The calibration curve and decision curve analysis (DCA) also showed that the radiomics model had an ideal discrimination power for LUAD (Figs. 6E, 6F).

## Evaluation of prognostic ability of radiomics model

The prognosis of LUAD samples in TCGA was also evaluated according to the radiomics model, generating Kaplan-Meier curves and ROC curves. Radiomics model could significantly distinguish the prognosis of different LUAD patients, and had the best prediction effect on 5-year overall survival, with ROC-AUC of 0.88 (Figs. 7A, 7B). The rad scores of samples exhibiting high TEDC2 expression were significantly higher than those of samples exhibiting low TEDC2 expression (Figs. 7C, 7D) in both the training and validation sets.

## Diagnostic markers were overexpressed in LUAD cells and promoted metastasis

We examined the expression of UBE2T, TEDC2, RCC1 and FAM136A in BEAS-2B and A549 cell lines, and found that they were all significantly overexpressed in A549 cells compared with BEAS-2B cells (Figs. 8A–8D). CCK-8 assays indicated that the proliferative capacity of A549 cells was significantly reduced after silencing TEDC2 expression (Fig. 8E). Compared with A549 cells without TEDC2 knockdown, the density of migrating and invading cells in either field was significantly loosed in A549 cells with TEDC2 knockdown, indicating that TEDC2 promotes the metastasis of LUAD cells (Figs. 8F, 8G).
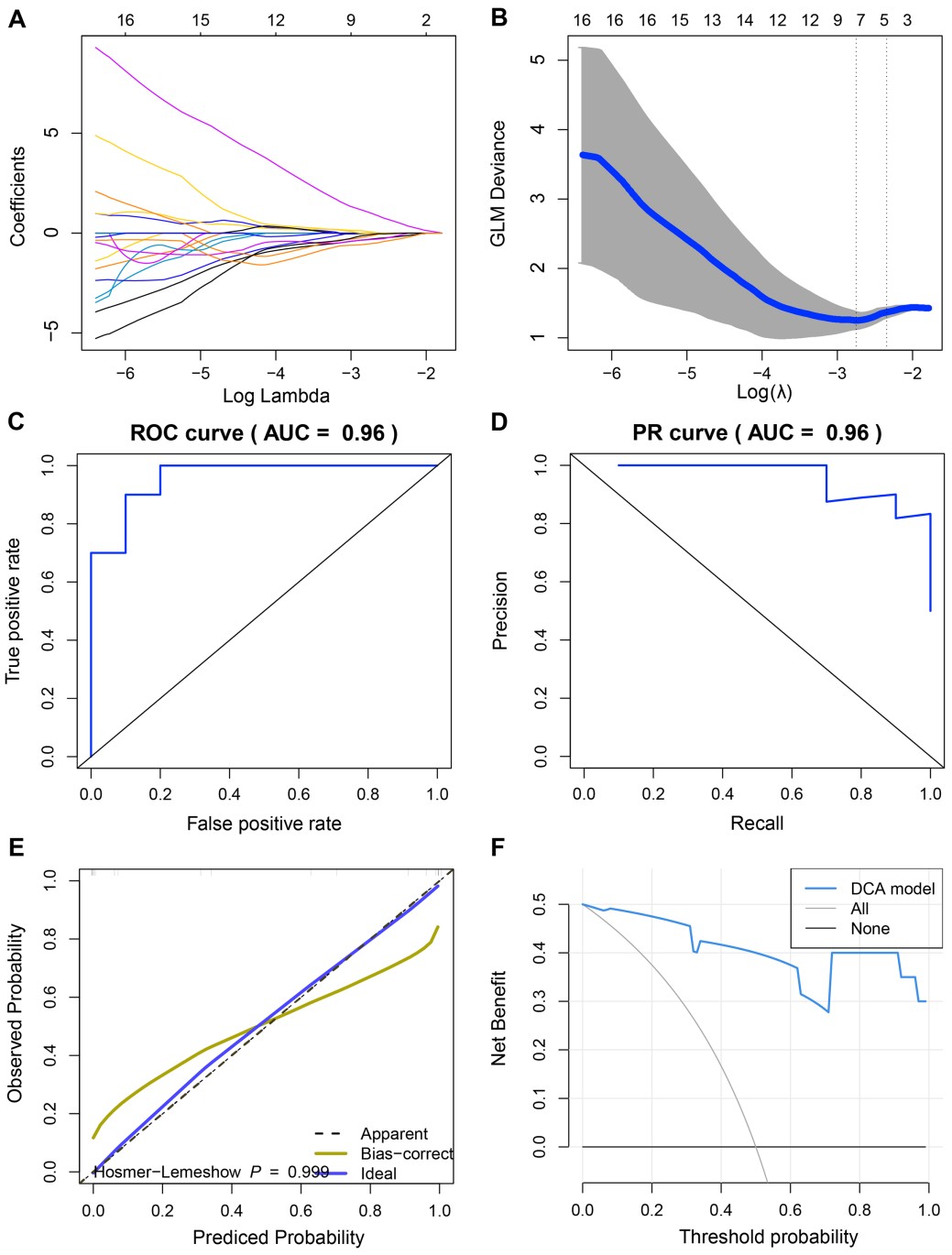

**Figure 6 Construction and discriminant ability evaluation of radiomics model.** (A and B) Radiomics features related to TEDC2 expression was selected by LASSO logistic regression analysis. (C) ROC curve of radiomics model in the training set. (D) PR curve of radiomics model in the validation set. The X-axis represents the actual positive rate (Recall), and the Y-axis represents the precision rate. The AUC-PR represents the average accuracy calculated for each coverage threshold. (E) Calibration curves for radiomics model. (F) DCA for radiomics model. The red curve is the radiomics model; the gray curve is the assumption that all patients were treated and the straight black line at the bottom of the figure is the assumption that no patients were treated.

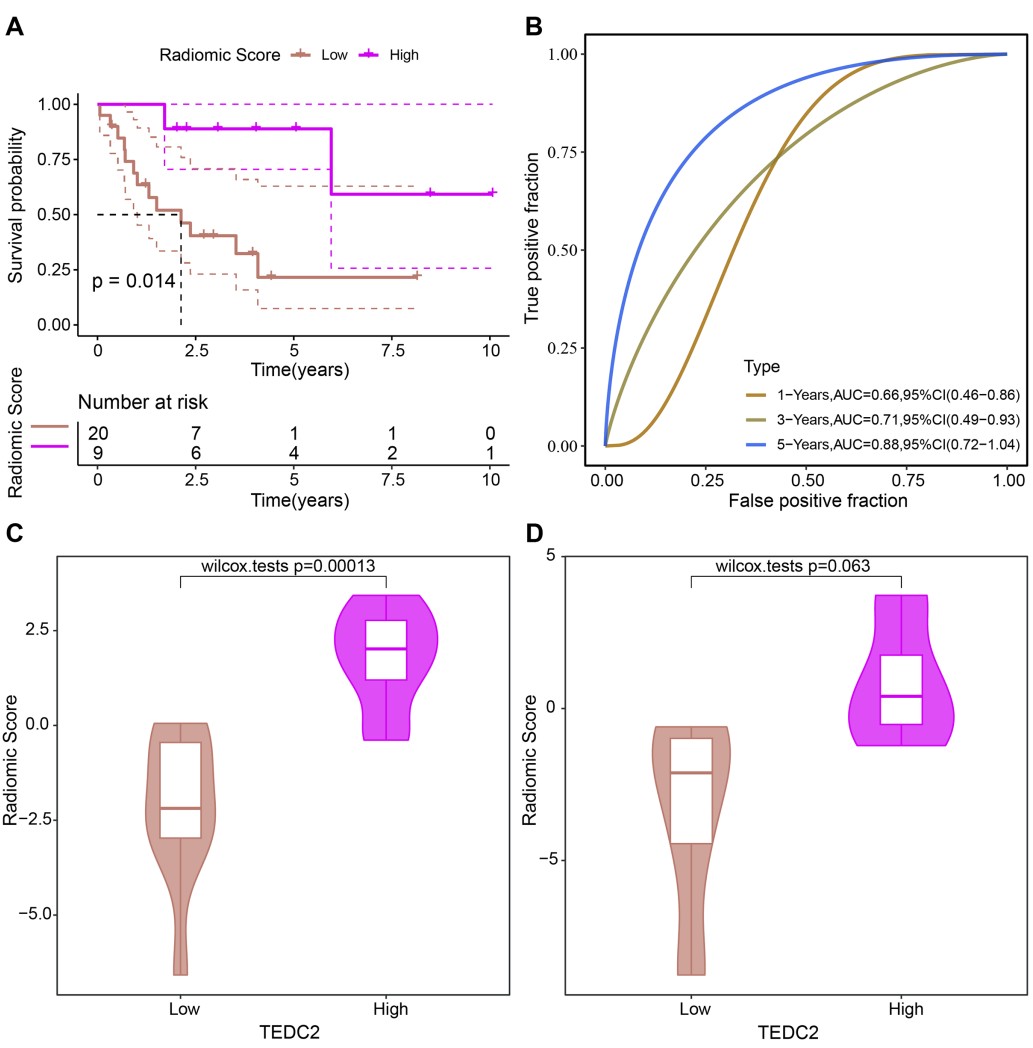

**Figure 7 Evaluation of prognostic ability of radiomics model.** Kaplan-Meier curves (A) and time-dependent ROC curves (B) generated by the radiomics model when evaluating the prognosis of samples from the TCGA-LUAD dataset. Rad score differences exhibited by high TEDC2 expression and low TEDC2 expression samples in the training (C) and validation (D) sets.

## DISCUSSION

Radiomics analysis is a quantitative approach to be applied precise diagnosis and treatment (*Lambin et al., 2017*). As radiomics is consolidated in translational cancer research and applied at the bedside, it is expected that radiomics data will be integrated and analyzed with genomics, proteomics, and other omics to provide valuable information for personalized medicine (*Limkin et al., 2017*). Machine learning is an area of current interest in medicine, particularly radiology, and may have a role in imaging-based screenings (*Ballard et al., 2021*). *Li et al. (2024)* constructed and validated a new combined radiomics and genomics model for predicting colorectal cancer metastasis by designing a multicenter, multiscale cohort. In addition, *Ye et al. (2024)* developed five radiomics-based machine learning models based on collecting information and extracting radiomics features from

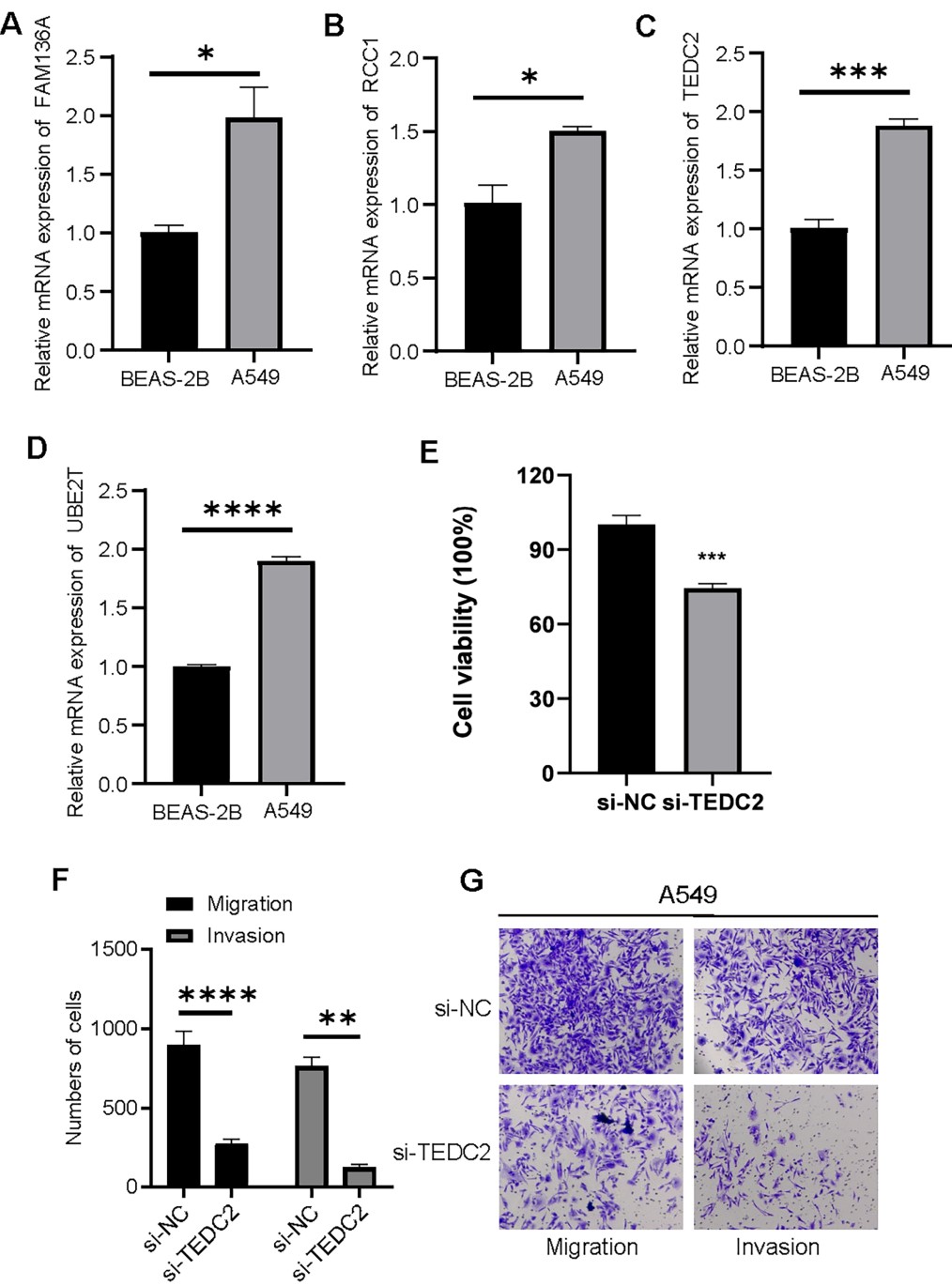

**Figure 8 Diagnostic markers were overexpressed in LUAD cells and promoted metastasis.** Relative mRNA levels of FAM136A(A), RCC1 (B), TEDC2 (C), UBE2T (D). CC8-K assay to determine the effect on cell proliferation after silencing TEDC2 expression (E). Transwell assay results of A549 cells with or without TEDC2 knockdown, (F) represents the number of migrated and invasive cells, (G) represents the image captured under the microscope. * represents $p < 0.05$, ** represents $p < 0.01$, *** represents $p < 0.001$, and **** represents $p < 0.0001$.

patients with pancreatic neuroendocrine tumors who underwent abdominal CT scans and developed five radiomics-based machine learning models. They found that the RF models based on interpretable radiomics can effectively distinguish between G1 and G2/3 of tumors, showing good interpretability. Thus, this study analyzed the genomics and radiomics data of LUAD and used machine learning methods to develop a diagnostic biomarker-based radiomics signature to provide markers with high specificity and sensitivity for the diagnosis of LUAD.

In oncology, biomarkers can be classified into several categories in terms of specific goals from predicting cancer susceptibility to prevention in clinical settings (*Bera et al., 2022*). Molecular tests relying on complex polygenic signatures are currently widely used in oncology (*Liang et al., 2024*). Different patterns based on data and genomic features will influence radiation oncology in the future (*Peeken, Nusslin & Combs, 2017*). At present, scientists investigate imaging biomarkers applicable for diagnosing and forecasting the pathological stage of non-small cell lung cancer by employing various machine learning techniques that rely on the analysis of CT image features (*Yu et al., 2019*). In addition, *Zhang et al. (2024)* used a volumetric CT-based radiomic signature to assess the tumor mutational burden (TMB) profile of preoperative LUAD patients and found that patients with high TMB all had significantly higher radiomic signatures than patients with low TMB. They concluded that a volumetric CT-based radiomic signature is beneficial for triage of LUAD patients for next-generation sequencing testing. In this study, the analysis to screen molecular markers from genomic data mainly consists of three parts: WGCNA, differential expression analysis, and three machine learning analyses. After these screening steps, we identified LUAD diagnostic signatures associated with radiomic features, including UBE2T, TEDC2, RCC1, and FAM136A.

A number of previous literatures have documented that UBE2T is highly expressed in lung cancer, which is diverse in molecular mechanism and has a carcinogenic effect in function, and is a prognostic risk factor for considerable types of malignant tumors such as lung cancer (*Yin et al., 2020*; *Zhu et al., 2021*; *Cao et al., 2022*). *Gao et al. (2021)* constructed a new radiogenomics biomarker based on a subgroup of hypoxia genes. They defined UBE2T as a hypoxia-associated genomic signature based on the TCGA database for renal clear cell carcinoma and demonstrated that the radiomic signature can be the best predictor of this gene in different cohorts (*Gao et al., 2021*). RCC1 functions critically in the regulation of cell cycle-related activities, and its upregulation is associated with adverse lung cancer prognosis, and manipulation of its expression in combination with PD-L1 antibody inhibits tumor growth in mice (*Zeng et al., 2021*). FAM136A activity is significantly increased in many lung cancer tissues and cells and is immunoreactive in the cytoplasm of lung cancer cells, where restriction of its expression exerts an inhibitory effect on essential components of tumorigenesis, including proliferation and metastasis (*Zhao et al., 2020*). A pure bioinformatics analysis study showed confirmed high-expressed TEDC2 as an independent LUAD prognostic factor. A large number of its co-expressed genes participate in the mitotic cell cycle process, and TEDC2 high expression indicated a low level of immune cell infiltration, particularly B cells and dendritic cells (*Fang et al., 2023*). Consistent with the findings of high expression of these genes in different cancer

types, our study found that TEDC2 and three other diagnostic markers were overexpressed in LUAD tissues and showed high diagnostic accuracy for LUAD. Notably, the negative correlations of four key markers, UBE2T, TEDC2, RCC1, and FAM136A, with stromal, immune, and ESTIMATE scores suggest that the high expression of these genes in LUAD may contribute to tumor growth and malignant behaviors by inhibiting the role of stromal and immune cells. These findings not only reveal the potential roles of these genes in tumor microenvironment regulation, but also are consistent with the existing knowledge of LUAD biology and provide important clues for further investigation of the functions and mechanisms of these genes.

However, we also need to recognize that this study has some limitations. First, the data in this study were mainly obtained from public databases, but due to the single source of data, they may be biased and cannot fully represent the heterogeneity of all LUAD patients. Therefore, future studies will introduce more sample data from different databases and regions to increase the size and diversity of the sample. In addition, although we screened for diagnostic markers, the specific mechanisms of these genes in the development of LUAD have not been explored in depth. This includes the use of multiple approaches such as animal models and gene editing techniques in order to explore their specific mechanisms in tumorigenesis and development. Finally, although radiomics models show high accuracy, their feasibility and cost-effectiveness in practical clinical applications have not been evaluated. We will continue to explore how radiomics modeling can be seamlessly integrated into existing clinical workflows to improve its practical application.

## CONCLUSION

We successfully screened a set of markers with high diagnostic clips for LUAD by innovatively combining transcriptomic and radiomic data. We also constructed a non-invasive diagnostic model based on radiomics signatures through the comprehensive analysis of machine learning algorithms. The expression level of TEDC2 was closely correlated with radiomic profiles, and a radiomic-based signature was constructed and validated based on it. In conclusion, this study provides initial insights and methods for the diagnosis of lung adenocarcinoma, but more in-depth research and validation are still necessary before applying them to clinical practice.

## ABBREVIATIONS

**CRC**        Colorectal cancer
**ScRNA-seq** single-cell RNA sequencing
**GEO**        Gene Expression Omnibus
**MMP**        Matrix Metalloprotease
**TAM**        tumor-associated macrophages
**PD-L1**      programmed death-ligand 1
**EMT**        epithelial-mesenchymal transformation
**CTLA-4**     Cytotoxic T lymphocyte antigen 4

### Funding

This study was funded by the 900th Hospital of the Joint Logistic Support Force of China: National Science and Technology Fund Incubation Special Program (No. 2023GK04) and the Fujian Province Young and Middle-aged Teacher Education Research Project (No. JAT210174). The funders had no role in study design, data collection and analysis, decision to publish, or preparation of the manuscript.

### Grant Disclosures

The following grant information was disclosed by the authors:
The 900th Hospital of the Joint Logistic Support Force of China: National Science and Technology Fund Incubation Special Program: 2023GK04.
Fujian Province Young and Middle-aged Teacher Education Research Project: JAT210174.

### Competing Interests

The authors declare that they have no competing interests.

### Author Contributions

- Qian Huang conceived and designed the experiments, analyzed the data, prepared figures and/or tables, and approved the final draft.
- Peng Zhang performed the experiments, analyzed the data, authored or reviewed drafts of the article, and approved the final draft.
- Zhixu Guo conceived and designed the experiments, prepared figures and/or tables, and approved the final draft.
- Min Li conceived and designed the experiments, analyzed the data, authored or reviewed drafts of the article, and approved the final draft.
- Chao Tao performed the experiments, prepared figures and/or tables, and approved the final draft.
- Zongyang Yu performed the experiments, authored or reviewed drafts of the article, and approved the final draft.

### Data Availability

The datasets generated during and/or analyzed during the current study are available at GSE: GSE30219 and GSE31210.

The raw data is available at GitHub and Zenodo:

- https://github.com/taochao1/Raw-data.git
- taochao1. (2024). taochao1/Raw-data: First release of my raw data (v1.1.0). Zenodo. https://doi.org/10.5281/zenodo.10784605.

https://www.ncbi.nlm.nih.gov/geo/query/acc.cgi?acc=GSE30219
https://www.ncbi.nlm.nih.gov/geo/query/acc.cgi?acc=GSE31210

## Supplemental Information

Supplemental information for this article can be found online at http://dx.doi.org/10.7717/peerj.18310#supplemental-information.

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
