# Peer review of "Comprehensive analysis of transcriptomics and radiomics revealed the potential of TEDC2 as a diagnostic marker for lung adenocarcinoma"

_PeerJ, doi:10.7717/peerj.18310_

## Round 0.1 · original submission · Major Revisions

While the reviewers acknowledge the novelty and comprehensiveness of your approach, they have raised several concerns regarding the methodology, data sources, and interpretation of results. Specifically, you need to provide more details on data preprocessing, feature selection algorithms, and the rationale behind your choices. Additionally, please expand on the functional analysis of the identified diagnostic genes, their clinical relevance, and potential limitations. We encourage you to thoroughly address the reviewers' comments and revise the manuscript accordingly. Upon receiving your revised manuscript, it will undergo another round of peer review to assess its suitability for publication. Please note that a point-by-point response to the reviewers' comments should accompany your revised manuscript.

Reviewer 1 ·

Basic reporting

see "Additional comments"

Experimental design

see "Additional comments"

Validity of the findings

see "Additional comments"

Additional comments

The theme of this study is to reveal diagnostic biomarkers for lung adenocarcinoma (LUAD) based on integrated transcriptomics and radiomics analysis. In this study, we first mined LUAD transcriptomic data and radiomic features through public databases. Then, WGCNA and other methods were utilized to screen diagnostic genes for LUAD. Finally, radiomic models were constructed by LASSO analysis to predict the association between the expression levels of diagnostic genes and cancer immunoregulation, which were validated by cellular experiments. The overall idea of this study is poor, but the following issues still need to be addressed before publication:
1. Why did this study not use the construction of PPI network and combined with WGCNA to screen the genes, is this traditional analysis method not applicable to this study? Or is the screening method in this study more advantageous?
2. Are the public databases involved in this study too homogeneous to validate the LUAD radiomics signature? Please provide a rational explanation.
3. Why did this study focus on the TME features of the model, and why did it not analyze these aspects of the model feature genes that are also important features for validating the biomedical properties of the genes in relation to the genomic mutation landscape, and drug susceptibility correlations?
4. Why did the cellular experiments carried out in this study focus on the link between the biomarkers and the cell migration and invasion phenotypes, without exploring their proliferative activity? Please give a rational explanation.
5. In Figure 1A, MEblue is indeed significantly positively correlated with module characteristics, but not to overlook MEbrown, which is most significantly negatively correlated with module characteristics, why not proceed to study this module? Is it that the module has no research value?
6. Is there any essential difference between the results of Figure 3 and Figure 4? Both results validate the clinical performance and accuracy of the 4 genes, is it possible to combine the two results?
7. It is recommended that the Introduction section describe more about the importance of screening biomarkers for the clinical diagnosis of LUAD, rather than simply stating the advantages and disadvantages of CT diagnosis for the identification of LUAD.
8. In the Introduction, it is recommended that more studies be added to describe what biomarkers are currently used for cancer radiomics characterization and what their shortcomings are.
9. This study was not clear in its description of limitations; please describe more clearly what the limitations of this study were, including whether the amount of data was adequate and whether follow-up studies included clinical trials.
10. Four biomarkers, UBE2T, TEDC2, RCC1 and FAM136A, were mined in this study, but the discussion only piled up the existing literature reports and did not reveal these four biomarkers in more depth in conjunction with the results of this study, and it is recommended to deepen the discussion in conjunction with the results of this study.

Reviewer 2 ·

Basic reporting

In the current study, the authors have investigated the gene signature of lung adenocarcinoma and associated imaging features. The topic of the study is interesting, and the overall study reads well. However, there are some major issues regarding the sections of the manuscript. In the introduction, I strongly suggest discussing any similar studies and their findings. The novelty of the current study should also be emphasized. In the materials section, you didn't provide enough details to capture the overall picture easily, which will lead to confusion and loss of attention for the readers. In the results, some key findings are not clear and need to be expressed. The limitations part should be moved to the discussion section.
Figure 2: Please revise the Figure 2C, you may use bar plot for clarity.
Figure 4: For ROC figures, the are should not be colored. Also, please consider revising the figure 4 A and D.
Figure 5: In the legend of B and D, you have presented three classes but only two classes were drawn.
Figure 7: Please revise the C and D. The center gap part may be confusing for the authors,.

Experimental design

There is confusion regarding the study design. At first, you have utilized different machine learning models to select the subset of genes, and later, you have utilized Lasso to generate the nomogram. Please clearly express the overall study framework.

Validity of the findings

The results are promising and presented with supporting figures.

Additional comments

Please see the specific comments for your manuscript

Abstract
1. Please extend the background section.
2. The methods section is not clear. Also, you didn't mention the evaluation techniques.

Main Text
1. Please revise the statement for OS.
2. The radiomics term was first utilized around 2012; however, the concept of texture analysis was performed for decades. Please revise the statement.
3. Please extend the detail for machine learning. How you optimize the hyper parameters for feature selection.
4. How did you decide which features to include? How did you compare the performance of the selected features. There seems to be a lack of presentation of the framework.
5. Sections 2.8 and 2.9 are confusing to follow. You performed secondary data analysis; therefore, the inclusion of these sections raises confusion.
6. Please fix the fix the formatting issue with the R2 value at line 166.
7. Please check line 183 for typos.
8. Please revise the following statement for clarity, "SVM-RFE selected 18 genes as feature numbers". Also, revise the similar statements for RF and LASSO.
9. Please revise the following statement for clarity, "The feature genes selected by the three machine learning algorithms coincided, and the coincident genes included"
10. After abbreviating, please directly use the "AUC" value within the manuscript.
11. The statement starting at line 210 is not complete, please revise it.
12. How did you select these 7 features?
13. Please clarify further feature selection to predict gene expression in section 3.6
14. How did you select these 7 features?

Reviewer 3 ·

Basic reporting

I appreciate the authors' efforts in combining radiomics and transcriptomics data to identify diagnostic markers for lung adenocarcinoma (LUAD) using machine learning techniques. The study presents a comprehensive approach involving data extraction, gene co-expression network analysis, and machine learning algorithms for gene selection, followed by experimental validation and construction of a radiomics model. However, there are several aspects that need clarification, expansion, or improvement to enhance the rigor, reproducibility, and impact of the research. Here are my detailed comments:

Data sources and preprocessing:
Specify the exact version of TCGA used, as updates might affect data availability and consistency.
Provide details on how radiomics features were extracted, including imaging modalities, segmentation methods, and the choice of radiomics software/platform.

Experimental design

Methodological clarity:
Explain the rationale behind choosing WGCNA, RF, LASSO, and SVM-RFE for gene selection. Discuss how these methods complement each other and contribute to the robustness of the identified diagnostic genes.
For each algorithm, report the specific parameters used, such as the number of trees in RF, regularization strength in LASSO, and kernel type/C value in SVM-RFE.

Validity of the findings

Gene validation and functional analysis:
Present the statistical significance (e.g., p-values, fold changes) of the differential expression analysis between LUAD and normal samples for UBE2T, TEDC2, RCC1, and FAM136A.
Elaborate on the negative correlation between the diagnostic genes' expression and stromal/immune scores. Discuss potential biological implications and how these findings align with existing knowledge about LUAD biology.

Additional comments

Clinical relevance and future perspectives:
Discuss the potential clinical utility of the identified diagnostic genes and the radiomics model, considering factors such as ease of measurement, cost-effectiveness, and potential integration into existing diagnostic workflows.
Address the limitations of the study, such as the retrospective nature of TCGA data, potential batch effects, and the need for external validation in independent cohorts.
Outline future research directions, including the investigation of therapeutic targets among the diagnostic genes, exploration of other machine learning techniques, and the extension of this approach to other lung cancer subtypes or malignancies.
In summary, the study presents a valuable contribution to the field by combining radiomics and transcriptomics data to identify diagnostic markers for LUAD. However, addressing the above points will strengthen the methodology, results interpretation, and overall impact of the research.

---

## Round 0.2 · Major Revisions

We have now received comments from three reviewers, two of whom have recommended acceptance with revisions, while one (Rev 2) has recommended rejection. After careful consideration of all the feedback, we have decided to invite you to submit a major revision of your manuscript. The reviewers have raised several important points that need to be addressed to improve the quality and impact of your work. In particular, please pay close attention to the following points:

1. Strengthen the background and rationale for your study, clearly articulating its novelty and contribution to the field.
2. Expand and clarify the methods section, providing more detail on your statistical analyses, model evaluation, and hyperparameter selection.
3. Enhance the presentation of results, including a more comprehensive discussion of your findings in the context of existing literature.
4. Address the concerns raised about the validity of your findings, considering the limitations of using a single dataset.
5. Include a thorough discussion of the study's limitations and potential future directions.
6. Revise the tone of your conclusions to better reflect the scope and design of your study.
7. Improve the abstract to more accurately and comprehensively represent your work.
8. Carefully review and update your citations to ensure accuracy and comprehensiveness.

Please provide a point-by-point response to all reviewer comments when submitting your revised manuscript. We look forward to receiving your revised manuscript.

Reviewer 1 ·

Basic reporting

no comment

Experimental design

no comment

Validity of the findings

no comment

Additional comments

The author proposed a prognostic model for lung cancer by combining transcriptomics and radiomics data, and validated it through experiments. The research is complete and the statistics are appropriate. I have no further comments

Reviewer 2 ·

Basic reporting

The topic of the study has previously been included in similar studies. As a secondary analysis of the public dataset, the findings of the recent study are pretty limited. The reasoning and hypothesis of the study is weak. Background was limited.

Experimental design

There are gaps in the analysis of the data. The methods section is very superficial and should be extended. The statistical analysis is not clearly described. The evaluation of the models is not mentioned.

Validity of the findings

The findings of the study are hard to validate due to the dependence on a single dataset. The patient characteristics were not included for clear dependence of the gene expression for the immune response. The results are indirectly associated with conclusions.

Additional comments

Please see the additional comments below

Abstract
1. Please revise the second statement of the methods section for a formal tone.
2. Which immune profiles were utilized?
3. Please mention the evaluation metrics in the methods section.
4. What were the criteria for selection of the diagnostic genes?
5. Which model resulted in the best model?
6. Please extend the results section. In its current form, it's very brief.
7. The conclusion statement needs to be revised in terms of tone. Based on the study design, it reads too strong.
8. Please revise the first statement of the background section.

Main Text
1. Please cite additional studies for the limitation of the therapeutic response mentioned in line 56.
2. Please correct the radiomics statement.
3. Please revise the following statement for correctness, " The concept of radiomics was first described in 2012". Quantitative imaging features have been used for medical image analysis for more than 3 decades. In a 2012 paper, the feature analysis procedure was named radiomics.
4. The lack of studies in oncology is also misleading. There are plenty of studies focusing on oncology data.
5. Please express the benefit of the current study. There are similar studies in the literature.
6. which approach did you use for the correction of the p values?
7. Please present the formula in the correct form.
8. Please do not use the function name in the manuscript, refer to the methods.
9. How did you select the hyperparameters?
10. There is a lack of information regarding the modeling, evaluation, and comparison.
11. The limitations of the study are missing.
12. Also, comparison of the current study with the literature is also missing.

Reviewer 3 ·

Basic reporting

The author utilized a multi omics analysis system to investigate the expression and potential molecular mechanisms of TEDC2 in lung adenocarcinoma. The references were timely, the structure of the article was clear, the charts were rich, and detailed raw data were provided

Experimental design

The idea of the manuscript is clear, innovative, and the experimental design is novel

Validity of the findings

Methodology provides detailed details, statistics are appropriate, and results and conclusions are well presented

Additional comments

no comment

---

## Round 0.3 · Minor Revisions

The reviewer has provided some additional minor comments and suggestions to further improve your paper. Please address the following points in your revision:

1. Refine the description of radiomics and its applications.
2. Provide more details about the databases used.
3. Proofread the article for any remaining typos.
4. Clarify the methods used for evaluating lab experiments.
5. Specify whether AUC values are for training or test sets.
6. Revise the phrase "normal one controls".
7. Explain the radiomics features used.
8. Include results of lab tests.
9. Moderate the strength of claims about the study's implications.
10. Revise the statement about radiomics and image characteristics.
11. Clarify the background of computer-assisted diagnosis approaches.
12. Remove function names from the manuscript text.

Please carefully consider these comments and make the necessary revisions. We look forward to receiving your updated manuscript.

Reviewer 2 ·

Basic reporting

The background is well expressed with sufficient literature.a

Experimental design

The experimental design is easy to follow.

Validity of the findings

THe study results are aligned with the conclusion.

Additional comments

1. Please revise the following statement for accuracy, “Radiomics, as a high-throughput method, has a wide range of applications in different aspects of the management of multiple cancers.”
2. Please briefly express details for the databases.
3. Please check the article for typos and correct them.
4. Please mention what methods you used for evaluation of the lab experiments.
5. Are AUC values for training or test sets?
6. Please revise the phrase “normal one controls”
7. Please mention what the rad features are.
8. Please share the results of the lab tests as well.
9. The following statement is too strong for the study, please soften it, “These findings not only deepen our understanding of LUAD biology, but also provide new ideas and approaches for clinical practice. ”
10. I appreciate your revision for the radiomics statement, however, the following is misleading “Radiomics, assesses image characteristics extensively and employs statistical techniques to pinpoint the features most closely linked to outcomes. ”
11. Please clarify the following “This approach is grounded in years of investigation into computer-assisted diagnosis and pattern detection ”
12. Please avoid using function name within the manuscript.

---

## Round 0.4 · accepted · Accept

After a thorough review of your responses, I am satisfied that you have comprehensively addressed all the reviewers' comments. The current version have provided detailed explanations for each point, made necessary revisions, and clarified aspects of the methodology and results as requested. The revisions have refined the description of radiomics, provided more details about the databases used, proofread the manuscript for typos, clarified their experimental methods, specified AUC values, revised problematic phrases, explained the radiomics features used, included lab test results, moderated the concerned claims, and addressed concerns about computer-assisted diagnosis approaches. Given the comprehensive nature of the revisions and your efforts in addressing each point, I do not think additional external review is necessary. I am confident that the current version is now suitable for publication without further external review.